# Subjective Happiness and Satisfaction in Postoperative Anisometropic Patients after Refractive Surgery for Myopia

**DOI:** 10.3390/jcm9113473

**Published:** 2020-10-28

**Authors:** Kazuno Negishi, Ikuko Toda, Masahiko Ayaki, Hidemasa Torii, Kazuo Tsubota

**Affiliations:** 1Department of Ophthalmology, Keio University School of Medicine, Tokyo 1608582, Japan; hidemasatorii@yahoo.co.jp (H.T.); tsubota@z3.keio.jp (K.T.); 2Minamiaoyama Eye Clinic, Tokyo 1070061, Japan; toda@minamiaoyama.or.jp; 3Otake Clinic Moon View Eye Center, Kanagawa 2420001, Japan; 4Tsubota Laboratory, Inc., Tokyo 1070061, Japan

**Keywords:** myopia, LASIK, happiness, satisfaction, anisometropia

## Abstract

Laser-assisted in situ keratomileusis (LASIK) contributes to increased patient happiness one month after surgery; however, longer term effects are unknown. We performed a retrospective cross-sectional study on 472 patients who underwent bilateral LASIK surgery to measure happiness and satisfaction with LASIK, and to identify affecting factors. Patients completed questionnaires on satisfaction with the surgery and the subjective happiness scale (SHS) before, and 1, 3, 6, and 12 months after surgery. Multiple regression analyses were performed to determine independent predictors of SHS and satisfaction scores. Mean SHS increased at one month but was similar to baseline levels by six months. The SHS of older patients was greater than younger ones at baseline and at one and three months, while satisfaction among the older group was poorer at one and three months. Multiple regression analyses revealed that the decrease in SHS score from one month to three months correlated with baseline SHS, SHS at one month, uncorrected distance visual acuity (UDVA), and age. Regression analysis revealed SHS at six months correlated with preoperative SHS, SHS at one month, and satisfaction at six months. Satisfaction at final visit correlated with age, UDVA, anisometropia, and with SHS at each visit. We conclude that happiness and satisfaction were age- and UDVA-dependent, and anisometropic patients report poorer satisfaction scores.

## 1. Introduction

Surgical correction of refractive errors restores uncorrected visual acuity and quality of life (QOL). Ophthalmic surgery drastically improves vision, even on the same day as the procedure, and patients experience the benefits of surgery. Cataract surgery is the most common surgical procedure for the elderly to improve vision and its effects on cognitive function, sleep and motor function [1,2,3]. Laser-assisted in situ keratomileusis (LASIK) for myopia correction is another established vision-restoring surgery, and beneficial for unaided vision and QOL in all generations [4,5,6,7]. Recent investigations described improved subjective happiness after LASIK [8] and cataract surgery [9] and a decline in dry eye symptoms [10]. Happiness is associated with health and disease, including longevity, QOL, cardiovascular diseases, and the neuroendocrine system [11,12,13,14]. Happiness or positive emotions are now regarded as a critical component of health [15,16,17], and subjective happiness can be measured with a validated questionnaire—the subjective happiness scale (SHS) [18,19].

Psychometric parameters may not remain stable for a long time, despite constantly normalized or improved physical status according to response shift theory [20]. Response shift theory has been applied to patient-reported outcomes when there are changes to the patients’ internal standards after the event. Likewise, happiness may not be stable after LASIK under constant ocular conditions in terms of visual acuity, refraction, and accommodation [21,22,23,24]. Despite the satisfactory surgical outcome of LASIK, such as spectacle independence, happiness may be autoregressive depending on individual life events [25,26] and happiness after LASIK has not been fully investigated.

Numerous repeats have associated satisfaction with LASIK and age, complications, and uncorrected distance visual acuity (UDVA) [21,22,23,24]. We hypothesized that anisometropia could be another factor contributing to the decline of satisfaction and happiness that may happen in some patients due to ocular conditions, surgical complications, and intentional monovision for presbyopic LASIK [27,28,29,30]. Anisometropia after LASIK could be a significant issue, both in unexpected and expected cases. One magnetic resonance imaging (MRI) study described cortical changes in anisometropic adults after LASIK, and the authors discussed improved fixational instability [31].

The aim of this study was to track changes in SHS and satisfaction after LASIK and explore predictors that affect psychological parameters. We focused on postoperative anisometropia, in addition to conventionally assessed age, refraction, visual acuity and presbyopia. The measurement of subjective happiness with the validated questionnaire “SHS” is a novel aspect of the current study.

## 2. Experimental Section

### 2.1. Patients and Ethical Approval

This study was a retrospective chart review of patients who underwent bilateral LASIK procedures at the Minamiaoyama Eye Clinic, Tokyo, between September 2011 and August 2014. Subjects completed preoperative and postoperative (1, 3, 6 and 12 months after surgery) questionnaires of SHS and satisfaction with surgery. The Institutional Review Board of the Minamiaoyama Eye Clinic approved the research protocol, and the study was conducted in accordance with the tenets of the Declaration of Helsinki. Informed consent was obtained with an opt-out option. 

### 2.2. Surgical Procedure and Ocular Examinations

Bilateral LASIK procedures were performed in succession on each patient using identical procedures. The corneal flap was created using an MK-2000 microkeratome (Nidek Co., Ltd., Aichi, Japan), an IntraLase FS60 (Abbott Medical Optics, Inc., Tokyo, Japan), or an IntraLase iFS laser (Abbott Medical Optics, Inc., Tokyo, Japan). Laser ablation was performed using the EC-5000 CXII excimer laser (Nidek Co., Ltd., Aichi, Japan). Detailed procedures, examinations, and postoperative medications have been described previously [8].

### 2.3. Outcome Measures

Outcome measures included UDVA, uncorrected near visual acuity (UNVA), manifest spherical and cylindrical powers, SHS, and satisfaction score. SHS was measured with the validated Japanese version of the SHS [19]. The scale is a four-item questionnaire of subjective happiness where each item requires patients to rate the statements on a seven-point Likert scale. Question and answers were: 1. “In general, I consider myself, (not a very happy person) 1-2-3-4-5-6-7 (a very happy person)”; 2. “Compared with most of my peers, I consider myself, (less happy) 1-2-3-4-5-6-7 (more happy)”; 3. “Some people are generally very happy. They enjoy life regardless of what is going on, getting the most out of everything. To what extent does this characterization describe you? (not at all) 1-2-3-4-5-6-7 (a great deal)”; 4. “Some people are generally not very happy. Although they are not depressed, they never seem as happy as they might be. To what extent does this characterization describe you? (not at all) 1-2-3-4-5-6-7 (a great deal)”. The overall SHS score was calculated by taking the mean of the responses of the four items after a rescaling was carried out for question 4. The possible scores ranged from one to seven and higher values corresponded to higher subjective happiness. A one-item questionnaire rated on a four-point Likert scale ranging from 1 (very satisfied), 2 (satisfied), 3 (less satisfied), to 4 (least satisfied) was used to measure patient satisfaction with surgery. A lower score indicated a higher level of satisfaction. Satisfaction score has been used in many studies with a Likert scale, visual analogue scale (VAS), and specific questions. For example, participants answered how strongly they agreed with the statement “I would recommend my current method of vision correction to a close friend or family members” for comparison of satisfaction between LASIK and contact lens prescription [22]. The SHS and satisfaction questionnaires were routinely employed for all patients scheduled for refractive surgery to aid in decision-making and we offered it at every visit before and after LASIK in our practice. However, some patients refused to complete the questionnaire, and sometimes appointments were cancelled after LASIK.

### 2.4. Statistical Analysis

The data obtained from the right eyes were used for all statistical analyses. Visual acuity was converted to the logarithm of the minimum angle of resolution (logMAR). Differences between the preoperative and postoperative SHS scores and satisfaction scores were tested using the Dunnett multiple comparison test. We then performed a multiple regression analysis to assess factors affecting SHS and patient satisfaction with LASIK surgery. Finally, we conducted a multiple regression analysis to investigate predictors of postoperative SHS, delta SHS (final SHS—SHS at one month), and possible predictors of satisfaction scores (SHS and satisfaction score at each visit, UDVA and UNVA at each visit, near add power, presence of anisometropia at one month after LASIK, sex, and age). Patients were stratified by age: <40 years of age (y) as the younger group, and ≥40 y as the older group. Anisometropia after LASIK was defined as anisometropia ≥ 0.75 D and/or UDVA in under-corrected (anisometropia ≥ 0.50 D) eye < 20/20. *p*-value < 0.05 was considered significant. All statistical analyses were performed using SPSS version 24 for Windows (SPSS Inc., Chicago, IL, USA).

## 3. Results

There were 472 participants (175 men, 37.1%) and mean age was 34.5 ± 9.7 y. Postoperative UDVA and refraction were stable up to six months (Table 1). Postoperative UDVA of ≥20/20 was achieved in 90.7% of participants, and a refractive error ≤ 0.5 D was achieved in 92.7% of participants. The number of returning participants was 331 at one month, 175 at three months, 123 at six months, and 34 at 12 months; therefore, we used data at 12 months for the results at final visit. The mean SHS of all participants increased one month after surgery (^†^
*p =* 0.002, vs. baseline, Dunnett test) and thereafter decreased to values similar to baseline at three months (*p =* 0.999) and at six months (*p =* 0.999), whereas satisfaction was unchanged at three and six months (Table 1, Figure 1).

Participants were next stratified by age according to a previous study [8]—342 patients were in the younger group (<40 y) and 130 in the older group (≥40 y). The number of patients in younger/older groups were 342/130 at baseline, 235/91 at one month, 119/54 at three months, and 94/23 at six months. The mean age of the younger/older groups was 29.7 ± 5.8 y/47.3 ± 5.4 y, and the preoperative refraction of the younger/older groups was −5.44 ± 2.41 D/−4.46 ± 2.91 D (*p <* 0.001). There was no difference in postoperative refraction and UDVA. The SHS scores of the older group were much greater than the younger group at baseline (*****
*p =* 0.005, unpaired *t*-test) and at one month (*****
*p <* 0.001), but decreased at three and six months, with no difference between the two groups, (*p =* 0.155 and *p =* 0.103, respectively). There was no significant change in SHS scores in the younger group (Figure 1). Satisfaction score was worse in the older group than the younger group at one (*p =* 0.003) and three months (*p =* 0.017), but not at six months (*p =* 0.128; Figure 1).

Multiple regression analysis revealed that the delta SHS ((SHS at final visit) − (SHS at 1 month)) correlated with SHS at one month (*p =* 0.018), preoperative SHS (*p =* 0.029), postoperative UDVA (*p =* 0.021), and age (*p =* 0.041; Table 2). Satisfaction with surgery at six months correlated with age (*p =* 0.047), preoperative SHS (*p =* 0.022), SHS at six months (*p =* 0.001), satisfaction at one month (*p <* 0.001), UDVA at six months (*p <* 0.001), and presence of isometropia (*p =* 0.020; Table 3). Preoperative and postoperative UNVA were not correlated with SHS or satisfaction scores.

Participants were next stratified by isometropia (*n* = 413, mean age 33.7 ± 9.5 y) and anisometropia (*n* = 60, mean age 40.3 ± 9.4 y). The mean anisometropia of the anisometropia group at final visit was 1.13 ± 0.53D in the monovision group and 0.58 ± 0.44 in the other anisometropia group (*p =* 0.367, unpaired *t*-test). The final satisfaction score of the isometropia/anisometropia group was one for 243 (59.7%)/15 (25.9%), two for 133 (32.7%)/27 (46.6%), three for 20 (4.9%)/12 (20.7%), and four for 1 (0.2%)/4 (6.9%; *p <* 0.001, Mann–Whitney U test; Figure 2). Satisfaction in the isometropic group was greater than in the anisometropic group at one month (*p =* 0.002, unpaired *t*-test), three months (*p <* 0.001), and six months (*p <* 0.001), whereas the SHS was similar between the two groups (Table 4). The final satisfaction score was 1.39 ± 0.58 in the younger isometropic group, 1.96 ± 0.92 in the younger anisometropic group, 1.79 ± 0.75 in the older isometropic group, and 2.19 ± 0.82 in the older anisometropic group (Figure 3). Among younger patients, there was a significant difference in satisfaction between isometropic and anisometropic groups (*p =* 0.002, Mann-Whitney U test, Bonferroni correction), but not among older patients (*p =* 0.098). SHS and satisfaction with surgery was similar between participants with monovision and anisometropia (Table 4).

## 4. Discussion

The present study successfully identified a subjective happiness increase at one month after LASIK using a validated questionnaire and then a subsequent decrease; this satisfaction remained stable after LASIK. Both psychometric parameters were age- and UDVA-dependent, and, in particular, satisfaction was poorer in participants with anisometropia. Multiple regression analysis revealed that the delta SHS correlated with age, preoperative SHS, satisfaction and UDVA, being consistent with a previous study for SHS at one month [8]. The current study indicates that patients with a greater improvement in SHS tended to regress to baseline, despite constant satisfaction. SHS and satisfaction were not correlated with UNVA, and this result confirmed that even older patients exclusively expected LASIK to be more useful for UDVA than UNVA.

We speculate that most of the younger patients without presbyopia fully enjoyed the benefits of improved UDVA and spectacle independence. In contrast, some of the older patients with decreased UNVA may struggle to adapt and accept the visual compromises inherent in presbyopia. This group also eventually need driving glasses and daily glasses, even after successful surgery due to declining visual function with age [27,28,29,30,31,32,33,34,35,36,37]. Older patients may also experience response shifts [20]; their SHS score was greater than that of the younger group at baseline, initially increasing, but then continuously decreasing until there was no difference with the younger group. Lensectomy with implantation of multifocal intraocular lens is sometimes a suitable option for the elderly with high myopia with early cataract [21]. Aged eyes may suffer dry eye symptoms, which is more prevalent in older populations [38,39], and a common pathology of decreased visual function with increased aberration and increased scattering originally induced by tear film instability on the ocular surface [40,41]. Dry eye symptoms may be newly developed after LASIK [42], and the loss of corneal innervation caused by flap making has been suggested as the major cause, affecting the corneal-lacrimal gland, corneal-blinking, and blinking meibomian gland reflexes, resulting in decreased aqueous and lipid tear secretion and mucin expression. Dry eye measurements, including the Schirmer test, conjunctivocorneal staining, tear film break-up time, as well as dry eye symptoms, should be evaluated in future studies.

Post-LASIK anisometropia can be classified as monovision and other anisometropia. Monovision is a planned anisometropia in which the dominant eye is corrected for distant vision, and the nondominant eye is corrected for near vision. Even after simulation with contact lenses, the acceptance by patients is unpredictable. The present study revealed satisfaction was poorer in anisometropic patients, even in cases of monovision, whereas SHS was similar in anisometropic and isometropic groups. We speculate that anisometropic patients may be happier with improved UDVA, and simultaneously disappointed by fixational instability and insufficient binocular vision. In particular, younger patients with anisometropia were dissatisfied at final visit compared with younger isometropic patients who may be sensitive to anisometropia or insufficient UDVA due to under- or overcorrection [31,43].

This study had several limitations. A questionnaire for reporting detailed QOL would be more helpful to assess participants’ happiness and satisfaction, as with the Refractive Status and Visual Profile (RSVF) [37], SF-36(MOS 36-Item Short-Form Health Survey) [44], and VFQ-25 (NEI Visual Function Questionnaire-25) [45], for example. Future studies would also benefit from including questionnaires for quality of life related to daily vision (such as night vision, vision for driving, vision for reading) to more comprehensively assess subjective happiness and satisfaction. Presbyopic examinations and dry eye examinations should be conducted for further evaluation of visual function in aged eyes. Finally, happiness should be linked to lifestyle-related parameters, and could be further confirmed with integrated regression analyses. A prospective study with a longer observation period is warranted to minimize drop-out cases and achieve more conclusive results for SHS after LASIK.

## 5. Conclusions

SHS was maximal one month after LASIK and thereafter regressed to baseline at three months. Satisfaction was stable between 1 and 12 months after LASIK. Both parameters were age- and UDVA-dependent, and younger anisometropic patients reported poorer satisfaction scores at their final visit. As is conventionally addressed, surgeons should be careful of indication and explanation for older patients. Special attention should be paid to postoperative younger anisometropic patients.

## Figures and Tables

**Figure 1 jcm-09-03473-f001:**
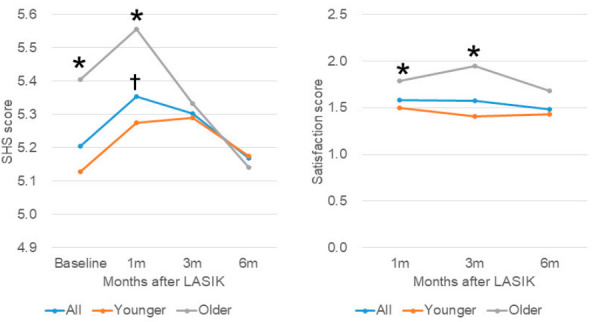
Subjective happiness score (left) and satisfaction scores (right) in patients stratified by age. (Left) Mean Subjective Happiness Score (SHS) of all participants (blue symbol) peaked at one month after surgery (^†^
*p =* 0.002, vs. baseline), but then decreased at three months (*p =* 0.999) and returned to baseline at six months (*p =* 0.999). SHS was greater in the older group (grey symbol) compared with the younger group (red symbol) at baseline (*****
*p =* 0.005) and at one month (*****
*p <* 0.001), but were similar at three months (*p =* 0.155) and at six months (*p =* 0.103). (Right) Mean satisfaction score of all participants (blue symbol) improved slightly but did not reach statistical significance. Satisfaction score was significantly worse in the older group (grey symbol) than the younger group (red symbol) at one month (*****
*p =* 0.003), at three months (*****
*p =* 0.017), but not at six months (*p =* 0.128).

**Figure 2 jcm-09-03473-f002:**
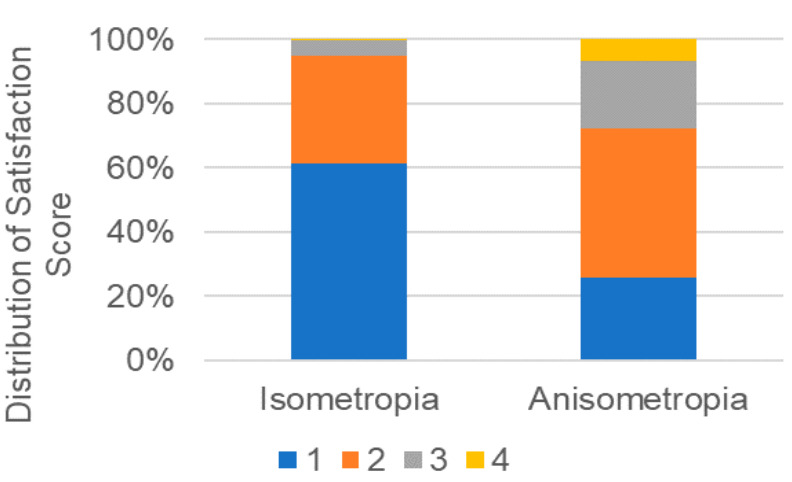
Distribution of final satisfaction score stratified by isometropia/anisometropia. Final satisfaction scores were significantly poorer in the anisometropia group (*p* < 0.001, Mann-Whitney U test).

**Figure 3 jcm-09-03473-f003:**
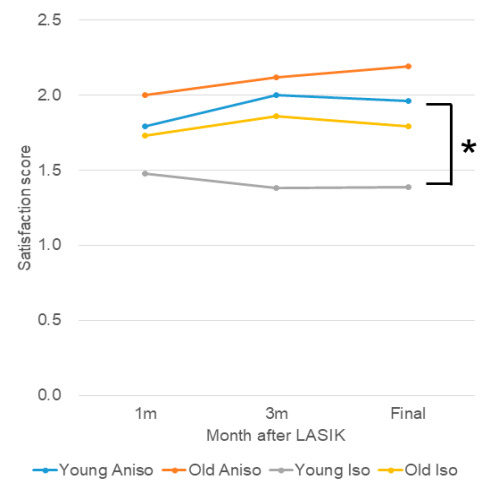
Satisfaction score stratified by anisometropia. The final satisfaction scores of the younger anisometropic group was poorer than the score of the younger isometropic group (*****
*p =* 0.002), whereas there was no difference between the older groups. Blue, younger anisotropic group; red, older anisometropic group; grey, younger isometropic group; yellow, older isometropic group.

**Table 1 jcm-09-03473-t001:** Ophthalmological and psychometric results.

	Time after LASIK			
	Baseline	1 m	3 m	6 m
Subjective Happiness Scale	5.20 ± 0.94	5.35 ± 0.94 (0.002)	5.30 ± 0.97 (0.999)	5.16 ± 1.01 (0.999)
Satisfaction		1.58 ± 0.66	1.57 ± 0.71 (0.999)	1.47 ± 0.62 (0.999)
Uncorrected Distance Visual Acuity	1.14 ± 0.31	−0.09 ± 0.15	−0.09 ± 0.15	−0.10 ± 0.13
Spherical equivalent (D)	−4.96 ± 2.32	0.02 ± 0.41	−0.03 ± 0.40	−0.06 ± 0.39
Cylindrical error (D)	0.84 ± 0.75	0.12 ± 0.29	0.11 ± 0.26	0.11 ± 0.27

*p*-value of multiple comparison in parentheses (Dunnett test vs. baseline for subjective happiness scale and vs. 1 month for satisfaction). Abbreviations: D, diopter; m, month(s).

**Table 2 jcm-09-03473-t002:** The results of linear and multiple regression analyses on factors associated with postoperative subjective happiness (SHS).

	Time after LASIK							
	1 m		3 m		6 m		DeltaSHS ^A^	
Independent Variables	β	P	β	P	β	P	β	P
Age	0.123	0.026	0.024	0.750	0.008	0.929	−0.166	0.051
	0.124	0.024 *	0.020	0.791	0.017	0.847	−0.176	0.041 *
Sex	−0.043	0.437	0.071	0.349	−0.125	0.178	0.041	0.629
	−0.047	0.391	0.070	0.359	−0.126	0.176	0.068	0.427
Preoperative SHS	0.749	<0.001 *	0.795	<0.001 *	0.460	<0.001 *	−0.217	0.010 *
	0.751	<0.001 *	0.706	<0.001 *	0.477	<0.001 *	−0.188	0.029 *
SHS at 1 m	−	−	0.795	<0.001 *	0.555	<0.001 *	−0.209	0.014 *
	−	−	0.771	<0.001 *	0.562	<0.001 *	−0.199	0.018 *
Satisfaction	−0.200	<0.001 *	−0.087	0.278	−0.234	0.011	−0.093	0.277
	−0.249	<0.001 *	−0.104	0.198	−0.242	0.001 *	0.032	0.722
Preoperative UDVA	−0.018	0.745	−0.188	0.013 *	0.104	0.264	0.067	0.428
	0.012	0.825	−0.196	0.010 *	0.119	0.208	0.031	0.714
Postoperative UDVA	−0.078	0.156	0.999	0.996	−0.121	0.191	−0.032	0.702
	−0.159	0.008 *	0.001	0.989	−0.142	0.141	−0.195	0.021 *
Preoperative UNVA	−0.059	0.292	0.100	0.193	−0.109	0.249	−0.092	0.289
	0.060	0.278	−0.118	0.128	0.085	0.380	0.029	0.741
Postoperative UNVA	−0.159	0.177	0.013	0.866	−0.241	0.266	0.197	0.242
	0.300	0.092	−0.003	0.978	0.386	0.189	0.170	0.504
Near add power	−0.097	0.477	−0.318	0.055	−0.003	0.991	−0.298	0.189
	−0.251	0.164	−0.394	0.106	−0.335	0.347	−0.322	0.229
Isometropia	0.005	0.928	0.004	0.957	−0.109	0.239	−0.013	0.876
	0.028	0.607	0.006	0.936	−0.108	0.253	−0.044	0.606

Dependent variable: postoperative SHS at each visit. *****
*p* < 0.05. The results of linear regression, upper rows; the results of multiple regression adjusted for age and sex, lower rows. men = 1, women = 0; isometropia = 1, anisometropia = 0. Delta SHS ^A^ = (value at final visit) − (value at 1 month). m, month(s); UDVA, uncorrected distance visual acuity; UNVA, uncorrected near visual acuity.

**Table 3 jcm-09-03473-t003:** The results of linear and multiple regression analyses on factors associated with satisfaction with the surgery.

	Time after LASIK					
	1 m		3 m		6 m	
Independent Variables	β	P	β	P	β	P
Age	0.268	<0.001 *	0.305	<0.001 *	0.180	0.045
	0.269	<0.001 *	0.304	<0.001 *	0.180	0.047 *
Sex	−0.017	0.746	0.045	0.551	0.023	0.794
	−0.029	0.583	0.028	0.699	0.009	0.920
Preoperative SHS	−0.160	0.003 *	0.026	0.726	−0.160	0.075
	−0.208	<0.001 *	−0.008	0.905	−0.209	0.022 *
SHS	−0.200	<0.001 *	−0.083	0.278	−0.234	0.011
	−0.234	<0.001 *	−0.095	0.198	−0.239	0.001 *
Satisfaction at 1 m	−	−	0.577	<0.001 *	0.451	<0.001 *
	−	−	0.555	<0.001 *	0.436	<0.001 *
Preoperative UDVA	−0.060	0.272	0.070	0.355	−0.059	0.524
	−0.004	0.928	0.0817	0.263	−0.029	0.749
Postoperative UDVA	0.445	<0.001 *	0.403	<0.001 *	0.368	<0.001 *
	0.409	<0.001 *	0.352	<0.001 *	0.350	<0.001 *
Preoperative UNVA	−0.062	0.259	−0.087	0.253	−0.005	0.956
	0.078	0.146	0.027	0.708	0.047	0.611
Postoperative UNVA	−0.366	0.001 *	−0.309	<0.001 *	−0.175	0.402
	−0.007	0.963	0.163	0.116	0.089	0.741
Near add power	0.192	0.158	0.101	0.549	−0.074	0.776
	−0.044	0.786	0.012	0.960	−0.119	0.743
Isometropia	−0.171	0.001 *	−0.284	<0.001 *	−0.302	<0.001 *
	−0.125	0.020 *	−0.218	0.004 *	−0.280	0.002 *

Dependent variable: satisfaction score at each visit. *****
*p* < 0.05. The results of linear regression, upper rows; the results of multiple regression adjusted for age and sex, lower rows. men = 1, women = 0; isometropia = 1, anisometropia = 0. m, month(s); SHS, subjective happiness scale; UDVA, uncorrected distance visual acuity; UNVA, uncorrected near visual acuity.

**Table 4 jcm-09-03473-t004:** Subjective happiness scale and satisfaction in isometropic and anisometropic participants.

	Time after LASIK			
**Subjective Happiness Scale**	**Baseline**	**1 m**	**3 m**	**6 m**
Isometropia	5.18 ± 0.95	5.35 ± 0.95	5.30 ± 0.96	5.12 ± 1.01
Anisometropia	5.37 ± 0.84	5.34 ± 0.80	5.29 ± 1.02	5.48 ± 0.98
*p*-value	0.140	0.928	0.957	0.239
**Satisfaction score**				
Isometropia	−	1.53 ± 0.62	1.49 ± 0.65	1.41 ± 0.54
Anisometropia	−	1.89 ± 0.83	2.08 ± 0.84	2.00 ± 0.87
*p*-value	−	0.002 *	<0.001*	<0.001 *
**Subjective Happiness**	**Baseline**	**1 m**	**3 m ****	**Final Visit**
Monovision (*n* = 10)	5.44 ± 0.88	6.05 ± 0.89	−	5.54 ± 1.06
Other anisometropia (*n* = 50)	5.39 ± 0.87	5.28 ± 0.80	−	5.27 ± 0.91
*p*-value	0.870	0.128	−	0.426
**Satisfaction**				
Monovision		1.80 ± 1.30	−	2.08 ± 0.90
Other anisometropia		1.91 ± 0.76	−	2.09 ± 0.86
*p*-value		0.863	−	0.990

******p* < 0.05, Unpaired t-test. The numbers of subjects are final visit. ****** No analysis due to small sample size. m, month(s).

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
