# Peer review of "Subjective Happiness and Satisfaction in Postoperative Anisometropic Patients after Refractive Surgery for Myopia"

_jcm, 2020, doi:10.3390/jcm9113473_

Round 1
Reviewer 1 Report
The manuscript has been improved including most of the suggested modifications.
I was not able to see Authors respences to reviewer's comments. How did the Authors deal with the query "Was the SHS routinely employed in all patients scheduled for refractive surgery? Why did the Author choose a retrospective design?"
Author Response
see file
Subjective happiness and satisfaction in postoperative anisometropic patients after refractive surgery for myopia
Response to The Reviewer:
Thank you very much for reviewing our manuscript. To aid in the review of the revised manuscript, we have included a point-by-point response to each comment.
Yours sincerely,
Kazuno Negishi, MD
Masahiko Ayaki, MD.
[Reviewer 1, comment ][I was not able to see Authors respences to reviewer's comments. How did the Authors deal with the query "Was the SHS routinely employed in all patients scheduled for refractive surgery? Why did the Author choose a retrospective design?"
We appreciate the reviewer’s comment. We already responded to the Reviewer’s comment in previous response letter; “The SHS was routinely employed for all patients scheduled for refractive surgery, and we offer it at every visit before and after LASIK in our practice. However, some patients refused to complete the questionnaire, and sometimes appointments were cancelled after LASIK. This was why we performed a retrospective chart review study” and now noticed confusing sentence in the Methods Section; “Written informed consent was obtained before LASIK surgery” despite this study was conducted as a retrospective chart review after surgery. We apologize for misleading sentence. We have modified the text as follows:
[Methods, page 2, line 78]” Informed consent was obtained with opt-out fashion. ”
[Methods, page 3, line 108] ”The SHS and satisfaction questionnaires were routinely employed for all patients scheduled for refractive surgery to aid a decision making, and we offer it at every visit before and after LASIK in our practice.”

Reviewer 2 Report
All my comments have been addressed.
The following change needs to be specified in the manuscript (currently it is added in the response but not changed in the manuscript).
Line 99: ''higher values corresponded to higher subjective happiness''. It needs to be stated that for question 4, a rescaling was done.
Author Response
see file
Subjective happiness and satisfaction in postoperative anisometropic patients after refractive surgery for myopia
Response to The Reviewer:
Thank you very much for reviewing our manuscript. To aid in the review of the revised manuscript, we have included a point-by-point response to each comment.
Yours sincerely,
Kazuno Negishi, MD
Masahiko Ayaki, MD.
[Reviewer 2, comment ][The following change needs to be specified in the manuscript (currently it is added in the response but not changed in the manuscript). Line 99: ''higher values corresponded to higher subjective happiness''. It needs to be stated that for question 4, a rescaling was done. ]
We appreciate the comment and added the sentences in the Method section.: [Methods, page3, line 99] ”The overall SHS score was calculated by taking the mean of the responses to the four items after a rescaling was done for question 4. The possible scores ranged from one to seven, and higher values corresponded to higher subjective happiness.”

This manuscript is a resubmission of an earlier submission. The following is a list of the peer review reports and author responses from that submission.
Round 1
Reviewer 1 Report
This is a human study to explore the subjective happiness scale (SHS) and satisfaction after LASIK, as well as the predictors that affect psychological parameters. They concluded that happiness and satisfaction were age- and uncorrected distance visual acuity-dependent, and anisometropic patients report poorer satisfaction scores.
- The dry eye measurement including the Schirmer test, conjunctivocorneal staining, the tear film break-up time, as well as the dry eye symptoms should be listed in the questionnaire.
- A questionnaire for quality of life related to daily vision (such as night vision, vision for driving, vision for reading) should be listed to assess happiness and satisfaction. Presbyopic status should be documented too.
Reviewer 2 Report
I think it is a well organized study and merits publication. I think it would be better if you commented on if patients were enhanced or had additional corrective surgery and how that influenced SHS scores.
Reviewer 3 Report
I would like to congratulate with the Authors for their work on this interesting topic. The paper is well written and has good readability.
Here are my suggestions
Introduction, page 2 line 63: please explicit MRI
Methods, page 3 lines 96-98: satisfaction questionnaire requires a more exhaustive explanation. Could you please insert the exact question of the item? Was it already employed in previous researches?
Results: please provided the results of univariate regression analysis
Figure1, line 149. The authors wrote "Satisfaction score was significantly worse in the older group (grey symbol)" but this is in contrast with the plot in which the grey line reaches higher scores. Please clarify
Was the SHS routinely employed in all patients scheduled for refractive surgery? Why did the Author choose a retrospective design?
I have some doubt regarding the novelty of the manuscript and its interest for the readers
Reviewer 4 Report
This manuscript by Negishi et al reports the relation between anisometropia and subjective happiness & satisfaction. There are few modifications needed to improve the clarity. The suggestions for modifications are given below:
- The title states anisometropic patients, the more exact expression would be ‘postoperative anisometropia’
- Abbreviation UDVA is used in abstract before defining
- Line 63: Emmetropization is not the correct term
- Line 65: It is not clear if it is preoperative or postoperative anisometropia
- Line 92-96: In Question 4, 7 corresponds to low subjective happiness. However, it is stated that higher values corresponded to higher subjective happiness. The overall score was calculated by taking the mean of the responses to the four items. Was there any rescaling done for question 4 before calculating the mean score?
- Statistical analysis: Only data from right eyes were included. What about the anisometropic eyes? UDVA is included in the multiple regression analysis. How does this address the worst eyes visual acuity?
- Table 1: What is the number of subjects included? Is this table based on 123 subjects (who returned for 6-month visit)?
- Line 127: What does the p value indicate here? If it is for the age, it is not necessary as the groups are divided based on the ages. It is unclear on how many subjects returned for later visits in each group.
- Table 4: Are the number of subjects stated for baseline visit or final visit?
- Line 218: ‘emmetropization’ is not the appropriate term
- In conclusion, it is stated that surgeons should be careful of indication and explanation for older patients. What is this statement based on?